# Cefiderocol Treatment for Severe Infections due to Difficult-to-Treat-Resistant Non-Fermentative Gram-Negative Bacilli in ICU Patients: A Case Series and Narrative Literature Review

**DOI:** 10.3390/antibiotics12060991

**Published:** 2023-06-01

**Authors:** Paul-Henri Wicky, Joséphine Poiraud, Manuel Alves, Juliette Patrier, Camille d’Humières, Minh Lê, Laura Kramer, Étienne de Montmollin, Laurent Massias, Laurence Armand-Lefèvre, Jean-François Timsit

**Affiliations:** 1Medical and Infectious Diseases Intensive Care Unit, AP-HP, Bichat Hospital, Paris Cité University, F-75018 Paris, France; paul-henri.wicky@aphp.fr (P.-H.W.); juliette.patrier@aphp.fr (J.P.); etienne.demontmollin@aphp.fr (É.d.M.); 2IAME INSERM UMR 1137, Paris Cité University, F-75018 Paris, France; josephine.poiraud@aphp.fr (J.P.); alves.manu@hotmail.fr (M.A.); camille.dhumieres@aphp.fr (C.d.); laurence.armand@aphp.fr (L.A.-L.); 3Bacteriology Laboratory, AP-HP, Bichat Hospital, Paris Cité University, F-75018 Paris, France; 4Pharmacology Department, AP-HP, Bichat Hospital, Paris Cité University, F-75018 Paris, France; 5Pharmacy, AP-HP, Bichat Hospital, Paris Cité University, F-75018 Paris, France

**Keywords:** cefiderocol, difficult-to-treat organisms, *Pseudomonas aeruginosa*, *Acinetobacter baumannii*, *Stenotrophomonas maltophilia*, critically ill, ECMO

## Abstract

Cefiderocol (FDC) is a siderophore cephalosporin now recognized as a new weapon in the treatment of difficult-to-treat-resistant (DTR) Gram-negative pathogens, including carbapenemase-producing enterobacterales and non-fermentative Gram-negative bacilli (GNB). This article reports our experience with an FDC-based regimen in the treatment of 16 extremely severe patients (invasive mechanical ventilation, 15/16; extracorporeal membrane oxygenation, 9/16; and renal replacement therapy, 8/16) infected with DTR GNB. Our case series provides detailed insight into the pharmacokinetic profile and the microbiological data in real-life conditions. In the narrative review, we discuss the interest of FDC in the treatment of non-fermentative GNB in critically ill patients. We reviewed the microbiological spectrum, resistance mechanisms, pharmacokinetics/pharmacodynamics, efficacy and safety profiles, and real-world evidence for FDC. On the basis of our experience and the available literature, we discuss the optimal FDC-based regimen, FDC dosage, and duration of therapy in critically ill patients with DTR non-fermentative GNB infections.

## 1. Introduction

Cefiderocol (FDC) is a novel class of cephalosporin, especially active against not only multi-drug-resistant (MDR) or extensively drug-resistant (XDR) Gram-negative bacilli (GNB), including carbapenemase-producing enterobacterales (CPE), but also carbapenem-resistant non-fermenting GNB, such as carbapenem-resistant *Pseudomonas aeruginosa* (CR-Pa), *Acinetobacter baumannii* (CR-Ab), and *Stenotrophomonas maltophilia* (Sm), also called difficult-to-treat-resistant (DTR) GNB [1]. FDC is a cephalosporin conjugated to a siderophore, which is a natural iron-chelating molecule used by bacteria to facilitate iron transport. Its activity is based on a “Trojan horse” strategy [2]. The FDC siderophore binds to free iron and enables the antibiotic to obtain active transport into the periplasmic space of Gram-negative bacteria via an iron-transporting outer membrane protein; in addition, it passively diffuses through porin channels, thus increasing its intracellular concentration. The antibiotic then binds to the penicillin-binding proteins (PBP), inhibiting the bacterial wall formation while blocking the production of the peptidoglycan [3]. 

### 1.1. Challenges for FDC Susceptibility Testing

Several methods have been developed to determine susceptibility to FDC, such as the agar diffusion disc method and minimal inhibitory concentrations (MICs) by E-test strips and by broth microdilution (BMD), but many discrepancies have been reported among the different methods, particularly due to the impact on the results of the iron concentration in the different media [4]. It is now recognized that iron-depleted cation-adjusted Mueller–Hinton media should be used. The European Committee for Antimicrobial Susceptibility Testing (EUCAST) continues to conduct works to elucidate the problems associated with FDC susceptibility testing [5]. In the meantime, MICs should be determined by broth microdilution in iron-depleted Mueller–Hinton broth [6]. 

EUCAST defined a susceptibility breakpoint at 2 mg/L for *P. aeruginosa* but not for *A. baumannii* and *S. maltophilia* due to insufficient clinical evidence to set reliable breakpoints. However, species-independent PK/PD critical concentrations with a breakpoint at 2 mg/L can be used for *A. baumannii, S. maltophilia*, and other species. The Clinical and Laboratory Standards Institute (CLSI) defined a susceptibility breakpoint of 4 mg/L, except for *S. maltophilia*, for which the breakpoint was changed to 1 mg/L in 2022 (Table 1) [7]. Susceptibility rates reported in epidemiological studies should be interpreted with caution, as the use of EUCAST or CLSI guidelines may alter the proportion of susceptible isolates [8,9].

### 1.2. In Vitro Activity 

FDC is active against Gram-negative bacteria, whereas it has no activity against Gram-positive bacteria or anaerobes. It is expected to be an adequate treatment not only against infections caused by MDR and XDR GNB, including non-fermenters, due to its high stability against all Ambler classes of beta-lactamases but also against the loss of porin channels and the upregulation of efflux pumps [11]. The in vitro activity of FDC has been evaluated in multi-national monitoring programs, SIDERO-WT (isolates from the United States and Europe) and SIDERO-CR (carbapenem-resistant isolates collected worldwide) [12]. *P. aeruginosa* showed a high susceptibility with MIC_90_ ≤ 2 mg/L, even on MDR and XDR isolates, leading to 99.8–100% susceptibility based on CLSI definitions. *S. maltophilia* isolates have high susceptibility rates ranging from 98.6 to 100%, with a low MIC_90_ (0.25 mg/L). *A. baumannii* have MIC_90_ ranging from 1 to 8 mg/L, resulting in the lowest susceptibility rate of 89.7% to 99.1%. Recently, the SENTRY antimicrobial surveillance program reported the in vitro activity of FDC against US and European non-fermenting clinical isolates [13]. For *P. aeruginosa*, FDC achieved 99.6% susceptibility against all isolates, according to CLSI criteria, and slightly lower susceptibility (97.3%) on XDR isolates, but it was still well above that of the newer antibiotics. *Acinetobacter* susceptibility to FDC was 97.7% and 95.8% for all *A. baumannii* and CR-Ab, respectively. *S. maltophilia* susceptibility to FDC was 97.9% [14] (Table 2).

### 1.3. Resistance

Resistance is uncommon in large multi-national cohorts, even among MDR and XDR isolates. In SIDERO-CR, 3.8% of isolates were not susceptible to FDC (according to CLSI breakpoints), but some series reported an alarming prevalence of resistance of up to 50%. The presence of some carbapenemase has been associated with resistance or reduced susceptibility to FDC [15]. The most frequent is the presence of New Delhi metallo-ß-lactamase (NDM), which resulted in a 4- to 16-fold increase in MIC when genes encoding NDM variants 1, 5, 7, or 9 were introduced into wild-type strains of *P. aeruginosa* or *A. baumannii* [16]. For clinical strains, a study reported that 54% and 27% of NDM-producing *P. aeruginosa* had MICs >2 mg/L and >4 mg/L, respectively, and that 50% and 20% of NDM-producing *A. baumannii* had MICs >2 mg/L and >4 mg/L, respectively [17]. 

Less commonly, the presence of a PER-type enzyme increased the MIC of *P. aeruginosa* from 2 to more than 32 times, depending on the variant. An English cohort reported 33% and 27% of PER-producing *P. aeruginosa* isolates had an FDC MIC >2 mg/L and >4 mg/L, respectively [18]. 

The emergence of mutants under treatment has also been observed. The different mechanisms include mutations in the chromosomal enzymes AmpC of *P. aeruginosa* or ADC of *A. baumannii*, which can increase the MIC up to 32 times, effecting the loss of membrane permeability (*oprD* mutation in *P. aeruginosa*) or overexpression of efflux pumps (MexAB and OprM in *P. aeruginosa* and smeDEF in *S. maltophilia*) [19]. Directly linked to its mechanism of action, resistance has been observed following mutations impacting siderophore receptors, the most frequent affecting the *piuA, piuD*, *and pirA* genes in *P. aeruginosa* or *A. baumannii* or affecting iron transport in *S. maltophilia* [20]. However, these mechanisms alone are insufficient for a strain to become resistant to FDC, and a combination of multiple resistance mechanisms is often required, the most frequent being a combination of β-lactamase production and permeability reduction. In a recent systematic review, Karakonstantis et al. examined the prevalence of resistance, including heteroresistance and resistance mechanisms in MDR and XDR GNB [21]. Heteroresistance to FDC appears to be common. It is defined by the presence of FDC-resistant subpopulations, although the methods for measuring heteroresistance are not clearly defined. The reported prevalence of FDC heteroresistance ranged from 10% in *P. aeruginosa*, 30% in *K. pneumoniae*, and 50% in *S. maltophilia* to almost 60% in *A. baumannii* [21,22]. The risk of heteroresistance is that resistant subpopulations may emerge during treatment, leading to treatment failure and the spread of resistant strains. The impact of heteroresistance (similar to colistin heteroresistance) on the clinical efficacy of FDCs has not been demonstrated.

### 1.4. Pharmacokinetic Properties

In Phase 1 clinical trials, the FDC t_1/2_ half-life elimination was mainly urinary, approximately 2–3 h in healthy subjects. The plasma protein-binding was 58%. Its tissue penetration and diffusion were linear and proportional to plasma exposure [23]. In patients with preserved kidney function (Glomerular Filtration Rate (GFR) between 90 and 120 mL/min) and those with mildly altered GFR (60 to 90 mL/min), the recommended dosage is 2 g intravenously every 8 h, administered upon a 3 h infusion, which should allow the concentration to reach more than 75% of the MIC in the interval between the two administrations. In the case of augmented renal clearance (GFR > 120 mL/min), an increased dosage is recommended, such as 2 g every 6 h. For patients with altered renal function, a lower dose is necessary. In the case of intermittent hemodialysis, the dose is decreased until reaching 750 mg q 12 h infused over 3 h. If a continuous renal replacement therapy (CRRT) is used, the dose is from 1.5 g q 12 h (if the effluent rate is ≤2 L/h) to 2 g q 8 h (if the effluent rate is ≥4.1 L/h) [24]. A monitoring of plasmatic concentration is useful to adapt the dosage, especially in patients under CRRT or extracorporeal membrane oxygenation (ECMO), to obtain an optimal trough concentration of 20 to 40 mg/L [25,26]. After multiple doses, FDC concentration in the epithelial lining fluid reaches 7.63 mg/L (8.93% of the total plasma concentration) at the end of the infusion and 10.4 mg/L (23.1% of the total plasma concentration) 2 h after the end of the perfusion [27].

The data regarding tolerance and safety suggest that the drug is well tolerated [28]. In the CREDIBLE-CR study, 30% of the patients treated with FDC had liver-related adverse events, most of those occurring in patients with previous liver diseases [29]. In addition, neurologic adverse events, such as seizures, were reported in only one patient, and there was no difference in the risk of *Clostridioides difficile* infection [30,31]

### 1.5. Previous Cohort Studies with FDC in Treating non-Fermentative GNB Infections in ICU

FDC use has been approved on the basis of the results of three randomized controlled trials performed in critically ill patients with nosocomial pneumonia, bloodstream infections, sepsis, or complicated urinary tract infections [29,30,31]. Importantly, in two out of the three studies, patients known to be infected with carbapenem-resistant GNB were not included. In the CREDIBLE-CR open-label randomized study, for infections due to carbapenem-resistant GNB, FDC use resulted in a clinical cure rate not inferior to the best available therapy (BAT). However, overall mortality rates were numerically higher in the FDC arm than in the BAT arm (34% vs. 18%, respectively, at end of therapy (EOT)). The higher mortality rate in FDC-treated patients was seen only among those with *A. baumannii* infections (FDC 50% and BAT 18% at end of study). Although this negative result in the *A. baumannii* subgroup might have been due to differences in the patients’ severity between FDC and BAT, a warning was given in the summary of product characteristics. That is why post-authorization real-life data are so important to consider, especially for infections due to *A. baumannii* and other non-fermentative GNB.

Two observational retrospective studies compared the FDC-based regimen to the colistin-based regimen for treating CR-Ab infections. Pascale et al. studied 107 patients (103 with SARS-CoV-2 infections) with bloodstream infections (BSI) (*n* = 62); skin and soft tissues infections (SSTI), ventilator-associated pneumonia (VAP), or hospital-acquired pneumonia (HAP) (*n* = 44); treated with FDC monotherapy (*n* = 42); or colistin-based therapy (*n* = 44). The authors showed similar risk of mortality (55% vs. 58%) and similar rates of clinical cure (40% vs. 36%) and adverse events [32]. Falcone et al. reported 124 CR-Ab severe infections (79 BSI, 35 VAP, and 8 other infections). Patients were admitted for SARS-CoV-2 pneumonia in 39% of the cases, 54% were invasively ventilated, and 60% were in septic shock. The 30-day mortality was lower in FDC-based regimens (15 monotherapy, 32 combination therapy, mortality 34%) than in colistin-based regimens (15 monotherapy, 62 combination therapy, mortality 55.8%, *p* = 0.018). This difference persisted after adjustment of the parameters imbalanced among strategies (propensity score adjusted HR 0.44; 95% CI [0.22–0.66], *p* < 0.001). The benefit of FDC was more important for BSI than for VAP. Importantly, FDC treatment failure was more frequent in monotherapy (6/14) than in combination therapy (2/32) (*p* = 0.006). Serious adverse events, mainly acute kidney injury, occurred more frequently in the colistin-based strategies [33]. 

The data regarding severe DTR-Pa infections are scarce. Gatti et al. published four cases of HAP/VAP or BSIs treated with an association of FDC and fosfomycin [34]. All four patients were alive at day 30 with microbiological cure. Gavaghan et al. reported six cases (three VAP, one BSI, and two SSTI) treated with FDC in monotherapy (four cases) or in combination therapy with aminoglycosides (two cases) [35]. Survival and clinical and microbiological successes were obtained in three cases (50%). Meschiari et al. reported 17 cases of DTR-Pa infections, including 14 treated with combination therapy. Pneumonia was the source of infection in 7 cases, intra-abdominal infections was the source in 4 cases; 15 patients were hospitalized in the ICU. Clinical cure was obtained in 12 (70.6%) cases, with clinical relapse in 3 cases. The 30-day all-cause mortality was 23.5% [36]. 

Data on therapeutic efficacy of FDC in severe infections due to *S. maltophilia* are very limited [35,37]. FDC might also be proposed as a last resort therapy, in combination with a second active agent, for severe *S. maltophilia* infections [38].

### 1.6. Case Series in ICU Patients at Bichat Hospital

We report here our experience in treating critically ill patients for severe infections due to DTR non-fermentative GNB with FDC between July 2020 and November 2021 in the Medical and Infectious Diseases ICU at Bichat Hospital, Paris, France.

## 2. Methods

### 2.1. Data Collection

Demographic and clinical characteristics were extracted in the ICU directly from the patients’ records and files. Results of bacteriological analysis were collected, with the type and date of sample, the strains isolated, and their susceptibility profile. 

Antimicrobial susceptibility testing was performed following the current EUCAST guidelines. When needed, MICs of relevant antibiotics were performed by the E-test or BMD method depending on EUCAST recommendations. MICs to FDC were determined by E-test or disk diffusion method and verified by ComASP^®^ Cefiderocol (Liofilchem) for all available isolates. Carbapenemase types were determined by Xpert^®^ Carba-R (Cepheid), and for some strains, whole genome was sequenced using Illumina technology.

The data concerning antimicrobial therapies focused on the dosage and route of administration and the duration of treatment. Concomitant medications and antimicrobials associated in combination with FDC administration were reported. The choice of the antibiotics was based on the MIC results. Dose regimens were as follows: tigecycline, 100 mg q12 h intravenously after a loading dose; colistin, intravenously 3 MUI q 8 h after a loading dose; and ciprofloxacin, 400 mg q 8 h intravenously. In the case of kidney failure, the dose of antibiotics was reassessed day-by-day according to the measured GFR and the need for renal replacement therapy (RRT), as usually recommended. Therapeutic drug monitoring (TDM) was performed whenever judged necessary, and trough plasma concentrations were determined using validated liquid chromatography coupled with tandem mass spectrometry [39]. Repeated samples were performed according to the patients’ clinical status (under hemodialysis, augmented renal clearance, etc.).

### 2.2. Definitions

The diagnosis of pneumonia relied on distal respiratory samples whenever judged relevant by the physicians if clinically suspected. We performed plugged telescopic catheter (PTC) and bronchoalveolar fluid leakage (BAL) and interpreted results using recommended thresholds for quantitative bacterial cultures as appropriate (10^3^ and 10^4^ CFU/mL, respectively).

A clinical cure was considered when the patients showed clinical signs of improvement, including decreased fever, resolution of sepsis or shock, recovery in organ dysfunction, and improved oxygenation persisting 2 days after EOT. A clinical failure was defined by an increased need for catecholamines, increase in FiO_2_ (%) or PEEP (cmH_2_0), worsening in organ dysfunction, or death, when occurring at least 2 days after the initiation of an adequate treatment based on antimicrobial susceptibility testing. 

Relapse was defined as a new infection with the same pathogen recovered from distal sampling (PTC or BAL) occurring more than 48 h after EOT. A superinfection was defined as a new infection due to a different pathogen occurring after completion of the previous course of treatment. Persistent colonization was considered when the same pathogen was found in a new respiratory sample, without signs of infection. Persistent colonization was not considered as a clinical failure and did not imply treatment course prolongation.

### 2.3. Outcome and Adverse Effects Assessment

The evaluation of adverse reactions under the FDC was based on clinical and biological findings. Hepatic toxicity was considered if liver enzymes (alanine aminotransferase, aspartate aminotransferase, bilirubin, and alkaline phosphatase) showed a 3-fold increase from baseline value within 48 h. Neurological toxicity was considered relevant if encephalopathy was clinically noticed at the bedside by the physician, namely delirium, agitation, seizures, or coma. Brain imaging and electroencephalography were routinely used to characterize abnormalities whenever deemed useful, at the discretion of the physician and ultimately to detect non-metabolic complications. Hypersensitivity was considered if a rash occurred and hypereosinophilia (>0.5 G/L) was observed. *C. difficile*-associated colitis was suspected in the case of secondary diarrhea.

## 3. Results

### 3.1. Patients’ Characteristics

We analysed 16 patients with DTR non-fermentative GNB infections in our ICU. Their main characteristics are shown in Table 3. Extracorporeal membrane oxygenation (ECMO) was used in nine (56.3%) patients. Eight patients (50%) required RRT during the treatment course, of whom four (25%) received continuous veno-venous hemodialysis (CVVHD) and four (25%) received intermittent hemodialysis (IHD). The source of infection was VAP in 14 (87.5%) cases. We reviewed 30 infectious episodes from 16 patients, CR-Ab in 9 (62.5%), XDR and CR-Pa in 7 (43.8%), *S. maltophilia* in 4 (25%), and carbapenemase-producing *Pseudomonas putida* in 1 (6.3%). 

VAP: ventilation-associated pneumonia; SSTI: skin and soft tissue infection; c-UTI: catheter-related urinary tract infection; CR: carbapenem-resistant; CR-Ab: carbapenem-resistant *Acinetobacter baumannii;* XDR-Pa: extensively resistant *Pseudomonas aeruginosa*. Results are presented as *n* (%) or median [IQR] for qualitative and quantitative variables respectively.

### 3.2. Clinical and Microbiological Outcomes

The median duration of FDC treatment was 8 days [7–13.5] (Table 4). Source control was conducted in three patients who had skin and soft tissues infection or urinary tract infection. In one case, since tissue debridement and antibiotic therapy were insufficient, leg amputation was decided due to treatment failure with CR-Ab infection. Of the 16 patients, 5 (31.3%) had clinical failure at EOT, and a relapse was observed in 9 patients (56.3%). Moreover, 13 (81.3%) patients had persistent colonization in the respiratory tract at EOT. The median length of ICU stay was 60.5 days [40–90.5]. Eleven patients were discharged from the ICU, and 10 were alive at hospital discharge and still alive after one year (Table 4).

The MIC for FDC was tested whenever possible throughout the relapses (Table 5). We observed no differences among pathogens. An increase from 4- to 8-fold in FDC MIC was observed in 3 patients with relapse of VAP due to CR-Ab. None had clinical failure at EOT due to decreased susceptibility while they were being treated with a combination therapy (tigecycline or colimycin). 

A combination therapy was used in 11 out of 30 episodes of infection (37%). Colistin was administered (nebulized or intravenously) in 7 episodes, whereas tigecyclin and ciprofloxacin was administered in 3 episodes (Table 5). In patients already under RRT, colistin was the preferred option. This was administered only for treating a relapse or a superinfection due to a second pathogen (except in Case 11) and for less than 7 days in all patients (data not shown). Colistin nebulization was used in four patients (Cases 1, 4, 13, and 16). In two patients, the last episode of VAP was treated with a tri-therapy, using ciprofloxacin and colistin. Case 1 had a VAP relapse due to *A. baumannii* and cefiderocol-resistant *P. aeruginosa.* In Case 16, a prolonged treatment course was decided (more than seven days) with nebulized colistin as a third antibiotic.

### 3.3. Adverse Events and Pharmacokinetic Profiles

The most frequent adverse events were hepatobiliary and neurological complications (Table 4 and Table 6). They were mostly characterized by mild cholestasis, with an elevation of less than 3-fold of bilirubin and alkaline phosphatase levels (data not shown). No clinically relevant cytolysis was observed in any patient. Discontinuation of treatment due to adverse events never occurred. Less frequent gastrointestinal complications, such as diarrhea (eight cases) and one *C. difficile*-associated colitis, were also observed. Fever was observed in two patients, which may have been related to FDC but did not require premature discontinuation. A hypersensitivity reaction was suspected in one patient (Case 6), who had a rash without hypereosinophilia, which was unlikely to be due to FDC (it appeared just before treatment initiation), and it disappeared after EOT. Neurological complications were observed in nine patients. As this occurred in sedated patients, the severity could not be clearly established, but it could have resulted in persistent hyporeactive delirium (Case 16). We did not observe seizures.

TDM was performed at least once in 12 patients (75%), 8 of whom were placed under RRT using either CVVHD (*n* = 4) or IHD (*n* = 4) (Table 6). The median time of FDC therapy before TDM was 3.5 days [2–5], and the median trough plasma concentration on the first sample was 34 mg/L [21–66]. Dose adjustments under RRT enabled therapeutic targets to be reached regardless of which CVVHD or IHD was used (Table 6). Two patients had glomerular hyperfiltration while being treated but showed trough plasma concentrations within the normal range. Of note, no association could be made between TDM and outcome. Moreover, neurological toxicity could not be statistically associated with elevated plasma concentrations.

**Table 5 antibiotics-12-00991-t005:** Cefiderocol Minimal Inhibitory Concentrations (MIC) and resistance mechanisms when identified for *A. baumannii*, *Pseudomonas sp*, and *Stenotrophomonas maltophilia* isolates.

			Microbiological Data						Outcome			
	Site	Combination Therapy	Isolate 1	MIC 1	Resistance Mechanism	Isolate 2	MIC 2	Resistance Mechanism	Clinical Cure	Persistent Colonization	Status at EOT	ICU Death
Case 1 ^§^	VAP_1_	No	*A. baumannii*	1	OXA-66 + ADC-30	-			Yes	Yes ^§^	Relapse	-
	VAP_2_	TG	*A. baumannii*	8	OXA-66+ ADC-30	*-*		-	Yes	Yes ^§^	Superinfection	-
	VAP_3_	COL	*A. baumannii*	8 *	OXA-66+ ADC-30-	*P. aeruginosa*	0.094 *	-	Yes	Yes	Relapse	-
	VAP_4_	COL/CF	*A. baumannii*	ND	OXA-66+ ADC-30-	*P. aeruginosa*	0.064 *	-	Yes	Yes	Relapse	No
Case 2	VAP	No	*S. maltophilia*	S **	-	-			Yes	No	None	No
Case 3	VAP	No	*S. maltophilia*	0.5 *	-	-			Yes	Yes	Relapse	Yes
Case 4	VAP_1_	No	*P. aeruginosa*	0.19 *	-	-			Yes	Yes	Relapse	-
	VAP_2_	No	*P. aeruginosa*	0.125 *					Yes	Yes	Superinfection	-
	VAP_3_	COL	*P. aeruginosa*	0.25	-	*A. baumannii*	2	ND	Yes	Yes	Superinfection	-
	VAP_4_	COL	*P. aeruginosa*	ND	-	*A. baumannii*	ND	ND	No	Yes	Relapse	Yes
Case 5	SSI	No	*P. aeruginosa*	0.5	VIM-2	-			Yes	No	None	No
Case 6	SSI	No	*P. aeruginosa*	<0.03	VIM-2	-			Yes	No	None	-
	UTI	No	*P. putida*	1	VIM-2	-			Yes	No	None	No
Case 7	VAP	No	*P. aeruginosa*	1	VIM-2	-			Yes	Yes	None	No
Case 8	VAP	No	*A. baumannii*	0.125	OXA-66+ ADC-30	-			Yes	Yes	None	No
Case 9 ^§^	VAP_1_	No	*A. baumannii*	0.125	OXA-66+ ADC-30	-			Yes	Yes	Relapse	-
	VAP_2_	No	*A. baumannii*	4	OXA-66+ ADC-30	-			Yes	Yes	Superinfection	-
	SSI	No	*A. baumannii*	1	OXA-66+ ADC-30	-			No	NA	None	No
Case 10	VAP		*A. baumannii*	0.25	OXA-66+ ADC-30	-			Yes	Yes	None	No
Case 11	VAP	TG	*A. baumannii*	0.5	OXA-58 + OXA-78 + VIM-4	-			Yes	Yes	None	No
Case 12	VAP	No	*S. maltophilia*	0.5	ND	-			No	Yes	Relapse	Yes
Case 13	VAP_1_	No	*A. baumannii*	0.5	OXA-66+ ADC-30	-			Yes	Yes	Relapse	-
	VAP_2_	TG	*A. baumannii*	S **	OXA-66+ ADC-30	-			Yes	Yes	None	No
Case 14 ^§^	VAP_1_	No	*A. baumannii*	S **	OXA-66+ ADC-30	*-*			Yes	Yes	Superinfection	-
	VAP_2_	CF	*A. baumannii*	0.25	OXA-66+ ADC-30	*P. aeruginosa*	0.25	ND	Yes	Yes ^§^	Relapse	No
	VAP_3_	COL	*A. baumannii*	1	OXA-66+ ADC-30	*P. aeruginosa*	S **	ND	Yes	Yes ^§^	None	No
Case 15	VAP	No	*S. maltophilia*	0.25	ND	-			No	Yes	Relapse	Yes
Case 16	VAP_1_	No	*A. baumannii*	0.5	OXA 23	-			Yes	Yes	Superinfection	-
	VAP_2_	COL	*A. baumannii*	0.38 *	OXA 23	*P. aeruginosa*	0.25	ND	Yes	Yes	Relapse	-
	VAP_3_	COL/CF	*A. baumannii*	0.5 *	OXA 23	*P aeruginosa*	0.25		No	Yes	Relapse	Yes

^§^ Patients with cefiderocol MIC increase; TG: Tigecyclin; COL: Colimycin; CF: Ciprofloxacin; NA: not available; ND: not determined. * MIC performed by E-test strip not verified by broth dilution method; S **: Strain susceptible to cefiderocol by disk diffusion method.

**Table 6 antibiotics-12-00991-t006:** Cefiderocol pharmacokinetic data during treatment for enrolled patients, FDC dosages, and potential related side effects.

	Site	Species	ECMO	Dosage 1				Dosage 2				Dosage 3				Dosage 4				Toxicity
				Interval	Dose	GFR	[C_trough_]	Interval	Dose	GFR	[C_trough_]	Interval	Dose	GFR	[C_trough_]	Interval	Dose	GFR	[C_trough_]	
Case 1	VAP_1_	*A. baumannii*	Yes	5	2000 q 8 h	CVVHD	51	6	2000 q 8 h	CVVHD	39	7	1000 q 12 h	CVVHD	39.5	9	1000 q 12 h	CVVHD	74	Encephalopathy
	VAP_2_	*A. baumannii*	Yes	-	-	-	NA	-	-	-	NA	-	-	-	NA	-	-	-	NA	-
	VAP_3_	*A. baumannii*	No	-	-	-	NA	-	-	-	NA	-	-	-	NA	-	-	-	NA	-
	VAP_4_	*A. baumannii*	No	-	-	-	NA	-	-	-	NA	-	-	-	NA	-	-	-	NA	-
Case 2	VAP	*S. maltophilia*	Yes	-	-	-	NA	-	-	-	NA	-	-	-	NA	-	-	-	NA	None
Case 3	VAP	*S. maltophilia*	No	-	-	-	NA	-	-	-	NA	-	-	-	NA	-	-	-	NA	Hepatitis
Case 4 ^§^	VAP_1_	*P. aeruginosa*	Yes	2	1000 q 12 h	IHD ^†^	66	2	1000 q 12 h	IHD ^††^	17.5	5	750 q 12 h	IHD	30	-	-	-	NA	Encephalopathy
	VAP_2_	*P. aeruginosa*	Yes	-	-	-	NA	-	-	-	NA	-	-	-	NA	-	-	-	NA	-
	VAP_3_	*P. aeruginosa*	Yes	-	-	-	NA	-	-	-	NA	-	-	-	NA	-	-	-	NA	-
	VAP_4_	*P. aeruginosa*	Yes	-	-	-	NA	-	-	-	NA	-	-	-	NA	-	-	-	NA	-
Case 5	SSI	*P. aeruginosa*	No	3	1000 q 8 h	42	64.5	5	1000 q 8 h	57	30	-	-	-	NA	-	-	-	NA	Hepatitis
Case 6	SSI	*P. aeruginosa*	No	1	2000 q 8 h	CVVHD	61.7	7	2000 q 8 h	CVVHD	42.1	-	-	-	NA	-	-	-	NA	Encephalopathy
Case 7	VAP	*P. aeruginosa*	Yes	5	2000 q 8 h	51	17	-	-	-	NA	-	-	-	NA	-	-	-	NA	None
Case 8 *	VAP	*A. baumannii*	Yes	4	2000 q 6 h	161	28	-	-	-	NA	-	-	-	NA	-	-	-	NA	None
Case 9	VAP_1_	*A. baumannii*	Yes	2	1000 q 12 h	IHD	62	4	1000 q 12 h	IHD	50	-	-	-	NA	-	-	-	NA	Hepatitis/ Encephalopathy
	VAP_2_	*A. baumannii*	Yes	-	-	-	NA	-	-	-	NA	-	-	-	NA	-	-	-	NA	-
	SSI	*A. baumannii*	No	-	-	-	NA	-	-	-	NA	-	-	-	NA	-	-	-	NA	-
Case 10	VAP	*A. baumannii*	Yes	5	2000 q 8 h	12	90	-	-	-	NA	-	-	-	NA	-	-	-	NA	Eosinophilia/ Encephalopathy
Case 11 ^§^	VAP	*A. baumannii*	No	3	750 q 12 h	IHD ^††^	21.9	3	750 q 12 h	IHD ^†^	77.8	3	750 q 12 h	IHD ^††^	42.7	-	-	-	NA	Encephalopathy
Case 12	VAP	*S. maltophilia*	No	-	-	-	NA	-	-	-	NA	-	-	-	NA	-	-	-	NA	None
Case 13 *	VAP_1_	*A. baumannii*	Yes	9	2000 q 8 h	281	21	-	-	-	NA	-	-	-	NA	-	-	-	NA	Encephalopathy
	VAP_2_	*A. baumannii*	Yes	-	-	-	NA	-	-	-	NA	-	-	-	NA	-	-	-	NA	-
Case 14	VAP_2_	*A. baumannii*	Yes	-	-	-	NA	-	-	-	NA	-	-	-	NA	-	-	-	NA	None
	VAP_3_	*A. baumannii*	Yes	4	1000 q 8 h	CVVHD	66	8	1000 q 8 h	CVVHD	58	10	1000 q 8 h	CVVHD	23	11	1000 q 8 h	CVVHD	51	-
Case 15	VAP	*S. maltophilia*	No	2	750 q 12 h	IHD	29	11	750 q 12 h	IHD	7.2	15	750 q 12 h	IHD	10.5	-	-	-	NA	Encephalopathy
Case 16	VAP_1_	*A. baumannii*	No	6	1500 q 8 h	62	11	-	-	-	NA	-	-	-	NA	-	-	-	NA	-
	VAP_2_	*A. baumannii*	No	1	1500 q 8 h	CVVHD	34	6	1500 q 8 h	CVVHD	36.5	16	750 q 12 h	IHD	50	19	750 q 12 h	IHD	32	Encephalopathy
	VAP_3_	*P. aeruginosa*	No	-	-	-	NA	-	-	-	NA	-	-	-	NA	-	-	-	NA	-

GFR: Glomerular filtration rate; CCVHD: continuous veno-veinous hemodialysis; IHD: intermittent hemodialysis; [C_trough_]: trough plasma concentrations; * Glomerular hyperfiltration; ^§^ Pharmacokinetic data fully available; ^†^ pre-dialysis dosage; ^††^ post-dialysis dosage. Intervals are expressed in days, dose is expressed in milligrams (mg) for each administration, indicating the frequency of administration (every 6 h, 8 h, or 12 h). GFR is expressed in mL/min (measures based on urine samples), [C_trough_] is expressed in mg/L. NA: not available.

## 4. Discussion

We report here the results of FDC treatment on a cohort of very severe ICU patients treated for severe late-onset nosocomial infections, mainly VAP, due to difficult-to-treat *A. baumannii*, *P. aeruginosa, S. maltophilia*, *and P. putida*. The conditions of the treated patients were extremely severe, with a median ICU stay of 60 days. Although clinical cure at EOT was approximately 70% (11/16), relapses and superinfections were common (9/11). Relapse required a new antimicrobial regimen including FDC. Patients survived for at least one year from hospital stay in 62.5% (10/16) of cases. Despite a large variation in kidney function between glomerular hyperfiltration and renal replacement therapy with or without ECMO, the trough concentrations were within the usual range, which suggests that the recommended doses were adapted to the most severe patients. 

Nevertheless, our data suggest that reliance on TDM remains paramount because of large intra- and interindividual variability to optimize therapy, especially in the most severe patients [25] under RRT and ECMO [40]. The high rate of relapse could be partially explained by the proportion of patients with SARS-CoV-2 acute respiratory failure and VAP. Relapses and superinfections were more frequently seen in those particular patients, with identified risk factors, such as being treated for non-fermentative GNB infections, requiring ECMO, and being treated with other broad-spectrum antimicrobials [41]. Despite acceptable plasma trough level, concentration obtained in the alveolar fluid may have been insufficient to eradicate the high inoculum of DTR non-fermentative GNB recovered from VAP episodes [26]. Indeed, the lung diffusion of FDC was approximately 20%, and it was likely even less in SARS-CoV-2 patients presenting severe alterations of microvascular lung perfusion [41].

The particularly high incidence of persistent colonization and relapse in our FDC-treated patients raises the question of the emergence of resistance mechanisms during treatment [42]. As described by some authors, there are concerns about the efficacy against non-fermenting GNB, with the emergence of resistance during treatment [14,20,43]. Heteroresistance to FDC, particularly in *A. baumannii*, has been found in vitro to be associated with bacterial regrowth after FDC exposure [22]. In our patients, relapse was not always associated with increased MICs. 

The toxicity of FDC has been widely assessed in randomized controlled trials and preclinical studies [28,44], with concerns about neurological and hepatobiliary disorders that deserve to be questioned. Indeed, we saw relatively frequent liver test abnormalities; however, FDC was maintained in all but one case. Neurotoxicity was suspected in nine cases, although a causal relationship could not be firmly established. Several confounding factors could be involved, such as hypoxemia, fever, inflammation, sepsis, postoperative period, and cardiac surgery. The lack of correlation between trough concentrations and the intensity of neurological signs did not allow us to consider a causal relationship between FDC and delirium or coma.

Considering the high rate of relapses/superinfections, FDC-based regimens should be optimized. FDC heteroresistance with frequent bacterial regrowth after FDC exposure presents a sound argument for combination therapy [22]. However, combination therapy with other drugs active in vitro is not yet recommended as a definite therapy by European [45] or US guidelines [46,47]. The use of bi- or tri-therapy is suggested so far as a rescue option. It was associated with less clinical failure in patients treated for CR-Ab infections, mostly BSIs, according to Falcone et al. [33]. Of note, the rational use of a combination of FDC and colimycin (intravenous or nebulized) needs to be explored in this population at higher risk of failure and mortality [32]. Potential synergism with fosfomycin has been described in CR-Ab strains and tested successfully in some case reports [48]. 

In our cases, relapses and superinfections were not influenced by the use of combination therapy or adjunctive nebulized colistin. A firm conclusion is limited by the small number of patients enrolled and requires further studies. The mean duration of therapy was 8 days, in accordance with the guidelines. The high rate of persistent colonization and relapse may suggest that the treatment was stopped too early. Indeed, in a randomized controlled trial, Bouglé et al. failed to demonstrate the non-inferiority of an 8-day antimicrobial regimen over a 15-day antibiotic regimen on mortality or recurrence in patients with VAP due to *P. aeruginosa* [49]. On the contrary, the systematic use of prolonged therapy in severe infections caused by DTR GNB may promote selection of FDC-resistant strains. The optimal duration of therapy remained a matter of controversy and should be decided on an individual basis. 

## 5. Conclusions

This narrative review and personal experience focused on the use of FDC-based regimens for the treatment of infections caused by DTR Gram-negative non-fermentative bacilli in critically ill patients. The use of FDC as a last resort option appeared to be safe in the most critically ill patients, with a high rate of clinical success and a good survival rate but a significant risk of relapses and superinfections. The dosing recommendations allow acceptable tolerability and acceptable concentrations to be achieved in routine practice, although the risk of neurotoxicity needs further investigation. Further data are needed on the optimal duration of therapy and the use of combination antibiotic therapy or adjunctive nebulized antibiotics. 

## Figures and Tables

**Table 1 antibiotics-12-00991-t001:** Cefiderocol breakpoints (2022) [5,10].

	CLSI Breakpoints (mg/L)			EUCAST Breakpoints (mg/L)	
	S	I	R	S	R
*P. aeruginosa*	≤4	8	≥16	≤2 *	>2
*A. baumannii*	≤4	8	≥16	≤2 *	>2
*S. maltophilia*	≤1	-	≥2	≤2 *	>2

S: susceptibility; I: Intermediate susceptibility; R: Resistant. * species-independent PK-PD breakpoint.

**Table 2 antibiotics-12-00991-t002:** Cumulative antimicrobial susceptibility testing results from SIDERO-WT and SIDERO-CR surveillance study isolates of *P. aeruginosa*, *A. baumannii* complex, and *S. maltophilia*.

	MIC_50_ (mg/L)	MIC_90_ (mg/L)	MIC Range (mg/L)
*P. aeruginosa*	0.12	0.5	≤0.002–8
MDR *P. aeruginosa*	-	2	0.002–32
*A. baumannii*	0.12–0.25	1–4	≤0.002–>256
MDR *A. baumannii*	-	2–8	0.015–>256
*S. maltophilia*	0.06–0.25	0.25–0.5	≤0.002–128
MDR *S. maltophilia*	-	0.25	0.015–>256

**Table 3 antibiotics-12-00991-t003:** Baseline characteristics at ICU admission and at cefiderocol initiation.

	All (*n* = 16)
Age	56.5 [52–66.8]
Gender	
Male	10 (62.5)
Female	6 (37.5)
Body mass index	27 [22–39]
Comorbidities	
Hypertension	11 (69)
Diabetes	7 (43.8)
Chronic kidney disease	4 (25)
COPD	1 (6.3)
Immunocompromised	2 (12.5)
ICU admission	
Cardiac surgery	6 (37.5)
SARS-CoV-2 pneumonia	8 (50)
Sepsis	1 (6.3)
Cardiac arrest	1 (6.3)
SOFA score	8 [3–13]
Albumin (g/L)	20 [18–22]
Treatment initiation	
SOFA score	10 [6–12]
Mechanical ventilation	15 (93.8)
Renal replacement therapy	8 (50)
Glomerular hyperfiltration	2 (12.5)
ECMO	9 (56.3)
Previous known colonization with CR pathogens	10 (62.5)
Site of infection	
VAP	14 (87.5)
SSTI	3 (18.8)
c-UTI	1 (6.3)
Pathogens	
CPE	-
CR-Ab	9 (56.3)
XDR-Pa	7 (43.8)
*P. putida*	1 (6.3)
*S. maltophilia*	4 (25)

**Table 4 antibiotics-12-00991-t004:** Main outcomes after treatment with FDC.

	All (*n* = 16)
Duration of antibiotic course (days)	8 [7–13.5]
Antibiotic association ^¶^	5 (31.3)
Source control ^¥^	3 (18.8)
Clinical failure	5 (31.3)
Persistent colonization	13 (81.3)
Relapse	9 (56.3)
Adverse events	
*C. difficile* colitis	1 (6.3)
Hepatitis	3 (18.8)
Eosinophilia	1 (6.3)
Encephalopathy	9 (56.3)
Rash	1 (6.3)
Discharged from ICU	11 (68.8)
ICU length of stay (days)	60.5 [40–90.5]
ICU mortality	5 (31.3)
In-hospital death	6 (37.5)
1-year death	6 (37.5)

^¶^ at least 2 or 3 antibiotics (administered intravenously or nebulized, see text for details); ^¥^ surgery or catheter removal. Results are presented as *n* (%) or median [IQR] for qualitative and quantitative variables, respectively. Abbreviations: FDC: Cefiderocol; ICU: intensive care unit.

## Data Availability

The datasets used and analyzed during the current study are available from the corresponding author on reasonable request.

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
