# Peer review of "Cefiderocol Treatment for Severe Infections due to Difficult-to-Treat-Resistant Non-Fermentative Gram-Negative Bacilli in ICU Patients: A Case Series and Narrative Literature Review"

_antibiotics, 2023, doi:10.3390/antibiotics12060991_

Round 1
Reviewer 1 Report
1. Table 1 has numerical values with an asterisk (*), but no table notes. Clarify it
2. Line 74. Use unified terms such as MDR, XDR, PDR. What does it mean `multi and highly drug resistant (MDR)`? What is `highly drug resistant`?
3. Line 79. ` in vitro` in italic
Author Response
- Table 1 has numerical values with an asterisk (*), but no table notes. Clarify it
Indeed, the legends were updated. we made the necessary changes in the manuscript to avoid confusion.
- Line 74. Use unified terms such as MDR, XDR, PDR. What does it mean `multi and highly drug resistant (MDR)`? What is `highly drug resistant`?
This point has been addressed and we made the necessary changes in the manuscript. Highly drug resistant has been deleted.
- Line 79. ` in vitro` in italic
The manuscript was modified as suggested.
Reviewer 2 Report
I have reviewed the manuscript entitled “Cefiderocol treatment for severe infections due to highly resistant non fermentative gram-negative bacteria in ICU patients: a case series and narrative literature review” submitted in “Antibiotics”. I found the manuscript in the scope of journal and the also the reader interest. The highlighted topic is of much importance in terms of antibiotic resistance and may provide good information to the scientific community. The authors have done great efforts in compiling all the relevant information in this review. In this manuscript, the authors have shared their personal experience and the data from a hospital, but they did not mention anything about the ethical approval from the institution. Although there is no direct interaction with the patients, but to my extent it is important to get the ethical approval from the institution as the authors are sharing the data. I don’t have any major comments on the manuscript. It can be proceeded further after addressing some minor changes.
My comments are:
1. The “gram-negative” should be changed as “Gram-negative” throughout the manuscript.
2. It is recommended to cite the references in table 1. (Ref for CLSI and EUCAST guidelines).
3. It is recommended to put a space between number and symbol throughout the manuscript e.g., at line 74, 131, 133: 2mg/L
4. It is recommended to italicize all of the bacterial and gene names throughout the manuscript e.g., A. baumannii at line 83.
5. Line 147, 155: Once the authors have out the abbreviated form for Acinetobacter baumannii, no need to put the full form later. Change it as A. baumannii.
6. The full form for CR-AB is already written at line 36. Please change it at line 155.
7. Line 171, 173: The authors are recommended to italicize the “et al” rather than author the names in in-text citations.
8. Lin 348: Change the P aeruginosa as “P. aeruginosa”.
9. The references needs to be updated according to the author guidelines for MDPI.
Good luck!
Author Response
My comments are: We thank the reviewer for all these suggestions, and we performed all suggested corrections in the manuscript.
- The “gram-negative” should be changed as “Gram-negative” throughout the manuscript.
Manuscript modified as suggested.
- It is recommended to cite the references in table 1. (Ref for CLSI and EUCAST guidelines).
Manuscript modified as suggested.
- It is recommended to put a space between number and symbol throughout the manuscript e.g., at line 74, 131, 133: 2mg/L
Manuscript modified as suggested.
- It is recommended to italicize all of the bacterial and gene names throughout the manuscript e.g., A. baumannii at line 83.
Manuscript modified as suggested.
- Line 147, 155: Once the authors have out the abbreviated form for Acinetobacter baumannii, no need to put the full form later. Change it as baumannii.
Manuscript modified as suggested.
- The full form for CR-AB is already written at line 36. Please change it at line 155.
Manuscript modified as suggested.
- Line 171, 173: The authors are recommended to italicize the “et al” rather than author the names in in-text citations.
Manuscript modified as suggested.
- Lin 348: Change the P aeruginosaas “ aeruginosa”.
Manuscript modified as suggested.
- The references needs to be updated according to the author guidelines for MDPI.
References updated as suggested.
Reviewer 3 Report
Reviewer comments
This manuscript provided a concise and well-structured narrative review of cefiderocol (FDC) treatment in severely infected patients at ICU. Beyond just the narrative literature review, this study also presented 16 severely infected clinical cases at ICU under the treatment of FDC and provided the clinical prospective of FDC experience. The manuscript is generally well-organized and scientifically sound. The contents fit the scope of the Journal. However, I have a few comments, both major and minor, to further improve the quality of the manuscript.
Major:
1. Table 3 and Table 4, it is confused about the number presented in these tables. There’s a mix of parathesis “()” and bracket “[]” with no annotation about what are the differences. Also, what is the meaning of “:” inside of the “[]”. Additionally, row #1, what is 56,5 means? Should that be 56.5? Please re-check the tables and reformat as needed to make it more clearer.
Minor:
1. MICs (page 2), SIDERO-CR (page 3), VAP, HAP, SSTI (page 4), CFD (page 5), VIM-2 (page 7), ARF, RCT (page 12) should have full name for its first appearance.
2. Table 1, please add table footnote for columns (“S”, “I”, “R”)
3. Page 2, “A. baumannii was the less susceptible species with MIC90s ranging from 1 to 8”, please add unit.
4. Page 5 “trough plasma concentrations were determined using validated liquid chromatography coupled with tandem mass spectrometry” please insert reference.
5. Table 6, should have units for interval, dose, GFR and [Ctrough] somewhere.
6. Page 8 “The most frequent adverse events were hepatobiliary and neurological complications 270 (Table 4).” It seems Table 4 doesn’t have the associated contents. Should be Table 6.
7. Page 12, “in the moost severe patients”, typo “moost”
Please check the above minor reviewer comments
Author Response
Major:
- Table 3 and Table 4, it is confused about the number presented in these tables. There’s a mix of parathesis “()” and bracket “[]” with no annotation about what are the differences. Also, what is the meaning of “:” inside of the “[]”. Additionally, row #1, what is 56,5 means? Should that be 56.5? Please re-check the tables and reformat as needed to make it clearer.
Thank you for this comment, we agree that this could be confusing. We modified the manuscript in order to clarify those informations in both tables, and let the annotations about the meaning down in the legend.
Minor:
- MICs (page 2), SIDERO-CR (page 3), VAP, HAP, SSTI (page 4), CFD (page 5), VIM-2 (page 7), ARF, RCT (page 12) should have full name for its first appearance.
We made some changes in the manuscript to improve clarity. Still, “SIDERO” is an acronym for a multinational surveillance program, for which the reference is indicated.
- Table 1, please add table footnote for columns (“S”, “I”, “R”).
We modified the manuscript as suggested
- Page 2, “A. baumannii was the less susceptible species with MIC90s ranging from 1 to 8”, please add unit.
We modified the manuscript as suggested
- Page 5 “trough plasma concentrations were determined using validated liquid chromatography coupled with tandem mass spectrometry” please insert reference.
References added.
- Table 6, should have units for interval, dose, GFR and [Ctrough] somewhere.
We made some change in the manuscript and footnotes of the table to make it clearer
- Page 8 “The most frequent adverse events were hepatobiliary and neurological complications 270 (Table 4).” It seems Table 4 doesn’t have the associated contents. Should be Table 6.
Manuscript corrected as suggested
- Page 12, “in the moost severe patients”, typo “moost”
Typo corrected.
Reviewer 4 Report

English language needs to be revised with the help of a native speaker. Also, there are several typos throughout the manuscript that need to be corrected.
Author Response
Furthermore, English language needs to be revised with the help of a native speaker.
done
Also, there are several typos throughout the manuscript that need to be corrected. In this form, I believe the manuscript should be extensively revised before resubmission.
Abstract Lines 18-19: “difficult-to-treat resistant (DTR) or extensively drug-resistant (XDR)” please choose DTT or MDR definition, avoiding merging them. Please add more information about the study part in the abstract which is too generic Introduction
We thank the reviewer for this comment about this semantic point, which had been corrected in the manuscript. Also, we added precisions in the introduction about the study part.
Line 50-51: “is necessary” I suggest changing this expression since it is misunderstandable
Thank you for your suggestion, manuscript modified as suggested.
Table 1: Please add a legend to better describe the asterisks and table’s meaning
Thank you for your suggestion, table modified and legend added.
Line 68: “Dedicated” I do not think this is the correct verb
Thank you for your suggestion, manuscript corrected
Lines 60 and 83-84: Please write bacteria names in italics (check the whole manuscript)
Thank you for your suggestion, manuscript modified as suggested
Lines 70-71: Please rephrase since English is incorrect Please write MIC90 in one way throughout the whole manuscript
Thank you for your suggestion, manuscript corrected
Line 112: Authors name should not be in italics Please read and discuss the following paper which fit with your point: 10.1128/spectrum.02347-22 Acinetobacter baumannii and Cefiderocol, between Cidality and Adaptability
The manuscript was modified as suggested. We thank the reviewer for suggesting this point and added some elements to the discussions. We agree to consider heteroresistance as an added argument in favor of combination therapy. We added this hypothesis at the end of the discussion section and the reference was added accordingly.
Line 124: As regards drug-drug interactions, I suggest being more specific
We thank the reviewer for this suggesting this point. As it does not add any value to the text, we decided to erase this sentence.
Line 130: “more than 75% of the MIC of the bacteria” what bacteria?
The manuscript was modified to improve clarity.
Line 134: “continuous technique” please use the appropriate term for CRRT
Manuscript modified as suggested.
Lines 137-138: “The lung penetration of FDC in mechanically ventilated patients with pneumonia was acceptable” I suggest avoiding this type of adjectives. You should report PK/PD information, without comments. Also, why do you use verb at past tense?
We thank the reviewer for suggesting this point. The manuscript was modified to improve clarity.
Lines 150-152: Please rephrase since it is unclear.
The manuscript was modified to improve clarity.
Lines 157-158: Check for repetitions “infections…infections”
The manuscript was reviewed and corrected as appropriate.
Please clarify what were the limits of the studies you reported. I suggest reading and discuss the following paper about cefiderocol and fosfomycin administration: 10.3390/antibiotics12010049
The observational nature of the studies was the most important limitation. The possible association with Fosfomycin is now added.
Methods 2 It is unclear what are the sample you analyzed to find bacteria. Did you perform blood cultures? Did you perform BAL? How did you decide if bacteria were pathogenic or colonization? This is a crucial point, please clarify.
We thank the reviewer for this crucial question, and we added information in the manuscript. Non-fermenting Gram-negative bacteria such as A. baumannii, P. aeruginosa and S. maltophilia were considered as pathogenic only if a clinical suspicion of pneumonia was present, especially on the first sample, in accordance with the European guidelines. Further superinfections and relapses were treated based on the same criteria for pneumonia, with a distal quantitative culture (PTC or BAL) at or above the threshold, whatever a persistent colonization was found or not. We did not observe any bloodstream infections due to those pathogens.
Results Why did you add information about colonization at EOT? Only infections should be treated. Please clarify this point in Methods section and Discussion section. It is a key point whether to treat A. baumannii or S. maltophilia. What antibiotics had been administered in combination therapy? And why? Also, add the dosage.
The molecules are now described within the table. The doses were adjusted on renal functions and are briefly described in the method section. We did not add further details because the table is already busy.
Please clarify. Please add more details about neurological adverse drug reactions.
We thank the reviewer for questioning this point. However, we unfortunately had only few possibilities to further explore those complications. We mentioned that we had mainly persisting coma as in Case 16, but no seizure. The brain dysfunction potentially associated with FDC was suspected when the patients showed lethargy or coma, unexplained by another condition. This is now briefly added in the manuscript.
Please talk about “Clostridioides difficile” as this is the new nomenclature.
The new nomenclature was applied. Providing further explanations on this nomenclature is beyond the scope of this manuscript.
Table 5 and 6: Please make simpler these tables, adding a legend to clarify Discussion
We are thankful for this remark, we tried to give a clearer and more structured view of the table containing the most relevant data, so it is more understandable and synthetic.
Line 298: FDC “administration” P Putida: Please add the full stop “P. putida” Please write SARS-CoV-2 in the correct form and check the whole manuscript for typos. The manuscript was reviewed and modified as suggested.
Line 335: “Combination therapy with other drugs active in vitro is not recommended”, I think there are not enough studies to say that. Please add more data and better discuss this point. In addition, please evaluate this paper about S. maltophilia VAP treatment: https://doi.org/10.3892/wasj.2023.193
We added references related to European and US guidelines and we added the experience with adjunctive Fosfomycin. We did not test this combination for our patients since most strains were resistant to Fosfomycin (P aeruginosa) or not tested
Round 2
Reviewer 3 Report
The authors adequately addressed my previous comments.